# CONTEXTREF: EVALUATING REFERENCELESS METRICS FOR IMAGE DESCRIPTION GENERATION

**Elisa Kreiss**[*]**, Eric Zelikman**[*]**, Christopher Potts, Nick Haber**
ekreiss@ucla.edu, {ezelikman, cgpotts, nhaber}@stanford.edu

## ABSTRACT

Referenceless metrics (e.g., CLIPScore) use pretrained vision–language models to assess image descriptions directly without costly ground-truth reference texts. Such methods can facilitate rapid progress, but only if they truly align with human preference judgments. In this paper, we introduce ContextRef, a benchmark for assessing referenceless metrics for such alignment. ContextRef has two components: human ratings along a variety of established quality dimensions, and ten diverse robustness checks designed to uncover fundamental weaknesses. A crucial aspect of ContextRef is that images and descriptions are presented in context, reflecting prior work showing that context is important for description quality. Using ContextRef, we assess a variety of pretrained models, scoring functions, and techniques for incorporating context. None of the methods is successful with ContextRef, but we show that careful fine-tuning yields substantial improvements. ContextRef remains a challenging benchmark though, in large part due to the challenge of context dependence.[1]

## 1 INTRODUCTION

Image description generation is an outstanding application area for image-based natural language generation (NLG). The purpose of an image description is to make the content of an image accessible to someone who can't see it. This most prominently affects people with temporary or long-term vision conditions, but it extends to people online facing image loading issues and those who simply prefer listening to PDFs and website content. Thus, the potential impact of work in this area is large.

In this context, recent proposals for referenceless evaluation metrics for image-based NLG are very welcome. Traditionally, evaluation in this area has been based on comparing a proposed description to a number of ground-truth descriptions (e.g, BLEU, Papineni et al. 2002; CIDEr, Vedantam et al. 2015; SPICE, Anderson et al. 2016; METEOR, Banerjee & Lavie 2005). Such *reference-based* metrics heavily rely on high-quality annotations (Anderson et al., 2016), which can be difficult to obtain. In contrast, referenceless metrics use pretrained vision–language models to assess image descriptions directly, without costly ground-truth reference texts. This serves a real-world need where ground-truth descriptions are sparse (Gleason et al., 2019; Williams et al., 2022; Kreiss et al., 2022b).

How well correlated are these referenceless metrics with human preferences, though? Unless there is a strong correlation, such metrics will lead us in wrong directions. To address this question, we present ContextRef, a new English-language benchmark for assessing referenceless metrics against human preferences. ContextRef has two components. The first derives from a human-subjects experiment eliciting ratings along a variety of quality dimensions (Figure 1A). The second provides ten diverse robustness checks designed to stress-test metrics via context manipulations, syntactically and semantically meaningful alterations to predicted texts, and changes to the input image (Figure 1B).

A crucial feature of ContextRef is that images and descriptions are presented in context. This reflects much recent work arguing that the context an image is presented in significantly shapes the appropriateness of a description (Stangl et al., 2020; 2021; Muehlbradt & Kane, 2022; Kreiss et al., 2022a). For instance, an image of a sculpture in a park presented in the context of a Wikipedia article on "Sculptures" will require a different description than when presented in an article on "Photographic Composition." In the first case, the sculpture and its properties should be prominent; in the second, the sculpture may require only a passing reference.

---

[*]These authors contributed equally to this work.

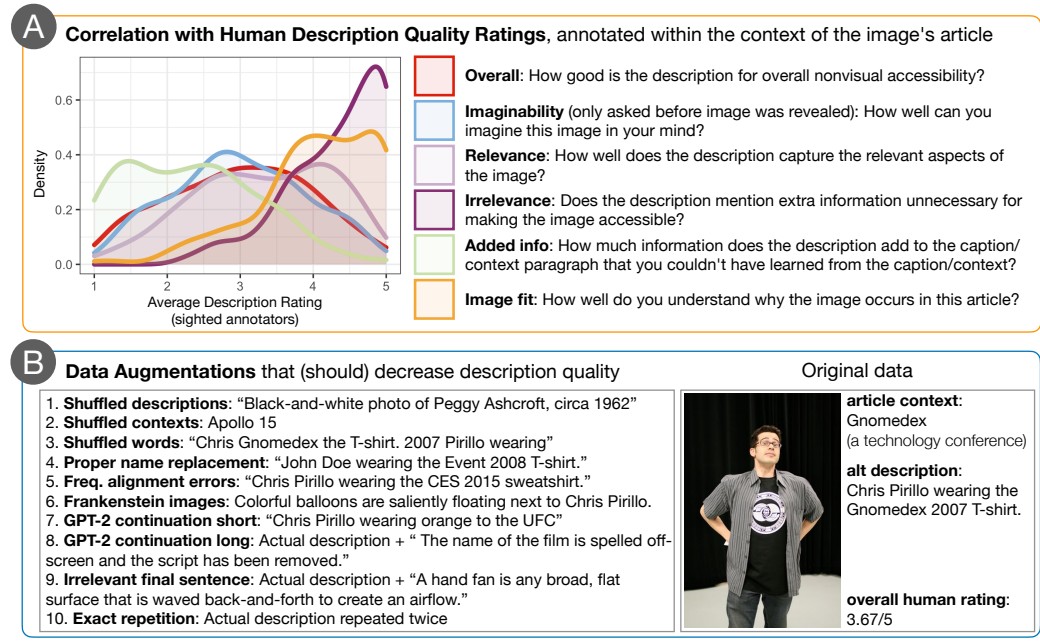

**A** **Correlation with Human Description Quality Ratings**, annotated within the context of the image's article

**Overall**: How good is the description for overall nonvisual accessibility?

**Imaginability** (only asked before image was revealed): How well can you imagine this image in your mind?

**Relevance**: How well does the description capture the relevant aspects of the image?

**Irrelevance**: Does the description mention extra information unnecessary for making the image accessible?

**Added info**: How much information does the description add to the caption/ context paragraph that you couldn't have learned from the caption/context?

**Image fit**: How well do you understand why the image occurs in this article?

**B** **Data Augmentations** that (should) decrease description quality

1. **Shuffled descriptions**: "Black-and-white photo of Peggy Ashcroft, circa 1962"
2. **Shuffled contexts**: Apollo 15
3. **Shuffled words**: "Chris Gnomedex the T-shirt. 2007 Pirillo wearing"
4. **Proper name replacement**: "John Doe wearing the Event 2008 T-shirt."
5. **Freq. alignment errors**: "Chris Pirillo wearing the CES 2015 sweatshirt."
6. **Frankenstein images**: Colorful balloons are saliently floating next to Chris Pirillo.
7. **GPT-2 continuation short**: "Chris Pirillo wearing orange to the UFC"
8. **GPT-2 continuation long**: Actual description + " The name of the film is spelled off-screen and the script has been removed."
9. **Irrelevant final sentence**: Actual description + "A hand fan is any broad, flat surface that is waved back-and-forth to create an airflow."
10. **Exact repetition**: Actual description repeated twice

Original data

**article context**:
Gnomedex
(a technology conference)

**alt description**:
Chris Pirillo wearing the
Gnomedex 2007 T-shirt.

**overall human rating**:
3.67/5

Figure 1: **Our proposed benchmark.** (A) ContextRef questions and distributions of averaged human ratings in the dataset for each question type. For simplicity, pre-image rating distributions are omitted (except for *imaginability* which only has pre-image ratings), since they show similar distribution patterns. Overall, the distributions are robust from the perspective of using the ratings to score reference-less metrics. (B) ContextRef example with illustrative robustness checks. These checks prove invaluable for uncovering undesired behavior of proposed metrics that can't be detected in naturalistic data.

We use ContextRef to assess a wide variety of referenceless metrics. The metrics we consider vary along three axes. First, we use a number of different pretrained models. Second, we consider two scoring methods: using the *similarity* of the learned image and description embeddings, and using the *likelihood* of the description conditioned on the image. Third, since prior referenceless metrics have not accounted for the role of context, we explore methods for integrating context into the metrics themselves.

None of the methods we explore succeed at ContextRef. In particular, while these methods mostly do show positive correlations with our human data, they fall short on our robustness checks, revealing that they are insensitive to fundamental changes to the examples they are evaluating. The main source of variation is the scoring method. In particular, similarity-based metrics tend to be less sensitive to grammaticality and context, while likelihood-based metrics tend to be less sensitive to uninformative but predictable text like repetition or irrelevant sentences.

However, we identify a path forward. Careful fine-tuning regimes can start making potential metrics much more successful at ContextRef. This is encouraging, but ContextRef remains a challenging benchmark. In particular, our fine-tuning experiments do not lead to models that are sufficiently sensitive to context, as reflected in ContextRef itself. However, we are optimistic that ContextRef can facilitate progress on this fundamental challenge for automatically generating useful image descriptions.

## 2  RELATED WORK

Referenceless metrics leverage pretrained vision–language models and provide scores for novel descriptions by considering the image directly (Hessel et al., 2021; Lee et al., 2021a;b; Scott et al., 2023; Lin et al., 2023)[2]. The most commonly used metric, CLIPScore (Hessel et al., 2021), assigns a score to each image–description pair based on the cosine similarity of the image and the description in CLIP's embedding space (Radford et al., 2021). CLIPScore often correlates better with human quality judgments than reference-based metrics (Hessel et al., 2021; Kasai et al., 2022), but its inability to integrate context significantly restricts its practical usefulness (Kreiss et al., 2022a). Kreiss et al. present initial evidence that context can be successfully integrated into the similarity computation of CLIPScore, and we develop this exploration much further (discussed in Section 3).

---

[2]See Appendix J for a more detailed overview on referenceless metrics.

In addition, recent vision–language models (many directly building on CLIP) have surpassed CLIP in downstream task performance on many multimodal tasks and offer new potential scoring opportunities. In this work, we investigate an array of models potentially capable of functioning as contextual metrics that leverage pretrained models, we investigate the role of similarity- vs. likelihood-based scoring, and we develop new methods for bringing in context.

An important feature of ContextRef is its series of robustness checks. Extensive research has been devoted to evaluating the robustness of models to input perturbations, especially in the context of adversarial attacks (Szegedy et al., 2014), including with multimodal models (Qiu et al., 2022; Kim et al., 2023; Pezzelle, 2023). In particular, works such as Ribeiro et al. (2020) highlight the value of leveraging interpretable changes to the input and confirming the model predictions change (or do not change) as expected. With ContextRef, we build on this work with a variety of previously-identified and novel robustness checks (see Section 5) to better understand the differences across scoring strategies.

## 3 MODELS AND SCORING STRATEGIES

In this section, we describe the models used for our experiments. For all of our approaches, the exact architectures of the visual and text encoders are designed to be easily interchangeable, and we tested many choices for each model. We selected current state-of-the-art vision-language models that cover a wide range of strategies for integrating textual and visual information, with varying degrees of multimodal pertaining. For consistency, we select one variant of each model according to their correlation with the human annotations and discuss the selected variants in Appendix D. We release the details for all models tested with the associated code. Based on the computation of the description quality score, we distinguish between likelihood-based and similarity-based metrics (similar to generative and discriminative scores in Lin et al. 2023).

### 3.1 LIKELIHOOD-BASED METRICS

Likelihood-based metrics score image descriptions conditional on the image and potentially other information like context. The precise method by which this is done depends on the model. To integrate context into these metrics without any fine-tuning, we considered two intuitive methods: (1) using the likelihood of a positive assessment of the description conditioned on an image description for an image and its context, and (2) using the likelihood of the description conditioned on a positive assessment, the image, and its context. We include the prompt templates used for the models in Appendix H, with all of these components. More precisely, we use the perplexity, $\ell(x) = \frac{1}{|x|} \sum_{i=1}^{|x|} \log \hat{p}_{LM}(x_i|x_{0:i-1})$, for a string $x$ with token length $|x|$ – however, we discuss explored alternatives in Appendix H. For method (2), we calculate the score as the perplexity where $x = \text{concatenate}(\text{image}, \text{context}, \text{description\_is\_high\_quality}, \text{actual\_description})$ – note that different models vary in how they take images as input, and we pass these in however the methods permit. Equivalently, in Python, this can be expressed with the simple template string `x = f"[Context : {context}] High quality, accessible, image description: {description}."`.

In initial experiments, it became clear that (2) is the superior option, so we focus on that method, as approach (1) peaked at about half of its correlational strength. There are multiple possible ways to calculate these scores; we found that using each language model's average per-token log-likelihood across the full sequence was consistently best correlated with human preferences across most models, as opposed to cumulative log-likelihood or only the log-likelihood of the conditioned variable.

**Flamingo** The OpenFlamingo v2 (Awadalla et al., 2023) models all use a CLIP-based image encoder (CLIP ViT-L/14), leveraging frozen, pretrained vision and language models. The visual features are passed into the language model using a cross-attention-based adapter. These models are a replication of the Flamingo work that introduced this cross-attention-based training method (Alayrac et al., 2022).

**Frozen** One approach to permit a text-only language model to operate as a multimodal model with no additional multimodal fine-tuning is to use a frozen language model (e.g., GPT-2; Radford et al. 2019) and a multimodal embedding model (e.g., CLIP; Radford et al. 2021) to map images to linear combinations of token embeddings. This combines ideas from Tsimpoukelli et al. (2021) and Norouzi et al. (2014) and was first introduced by dzryk (2023). We include more intuition in Appendix H.2.

**BLIP** The BLIP models that we consider (more precisely, BLIP-2 models; Li et al. 2023) use a ViT image encoder (Dosovitskiy et al., 2021), similar to the Flamingo models. Both OpenFlamingo

and BLIP support a variety of Transformer-based autoregressive text encoders, some of which are instruction-tuned (including InstructBLIP, which is instruction-tuned to follow directions; Dai et al. 2023). Unlike the other models, they are trained with both a likelihood-based and similarity-based objective. We analyze both their likelihood-based and similarity-based metric outputs.

## 3.2 Similarity-based Metrics

**CLIP** CLIP is a widely used multimodal technique mapping text and images to a shared embedding space using a contrastive objective (i.e., bringing together the embeddings associated with ground-truth text–image pairs while moving apart unassociated text-image pairs; Radford et al. 2021). Trained on large amounts of data, CLIP-based methods for image description evaluation (in particular, CLIPScore; Hessel et al. 2021) have been proposed.

We can incorporate context by including terms that take into account the cosine similarity between the context and the image or between the description and the context. We use the method proposed in Kreiss et al. (2022a), which shows a promising correlation with sighted as well as blind and low vision participant quality judgments. Intuitively, the method amends CLIPScore to incorporate the similarity of the description and context and replaces the similarity of the description to the image with the similarity of the description to information added by the image to the context. Explicitly, this is $\overline{\text{description}} \cdot \overline{\text{context}} + \overline{\text{description}} \cdot (\overline{\text{image}} - \overline{\text{context}})$, where $\overline{x} \triangleq \frac{x}{|x|}$. We use this as our main CLIP method and refer to the original CLIPScore (i.e., $\overline{\text{image}} \cdot \overline{\text{description}}$) as *Orig. CLIPScore*.

However, despite their widespread use, CLIP-based approaches generally suffer some key limitations. First, the most widely used Vision Transformer (ViT) models (but not ResNet models; He et al. 2016) expect center-cropped images, which fundamentally limits their usefulness as image-description-evaluation tools. In addition, for the default text encoder for CLIP, there is a 77-token character limit, which also applies to the substantial majority of the text encoders in OpenCLIP (note, however, that this doesn't apply to all of the text encoders in OpenCLIP, e.g., to RoBERTA; Ilharco et al. 2021). We also include CoCa under this umbrella, which modifies CLIP by adding an additional image captioning objective to the language model and is included in OpenCLIP (Yu et al., 2022b).

**BLIP** As mentioned, BLIP is trained with both likelihood and similarity objectives. Consequently, we evaluate both objectives in this study. Notably, BLIP is actually trained with two similarity objectives – an item matching and an item contrastive score – but, in this study, we focus on the item contrastive score since it tended to achieve higher correlation with our human judgment data. To compute the description quality scores, we use BLIP embeddings in the same way we use CLIP embeddings.

## 4 ContextRef: Evaluating Correlation with Human Judgments

The first part of ContextRef allows users to correlate model-assigned scores with human preference ratings. Image description quality judgments have been extensively studied; Bernardi et al. (2016) provide an overview of the various dimensions prior research has explored for determining quality, including accuracy grammaticality, creativity, and human-like content, which we further elaborate on in Appendix J. More recent frameworks include THumB (Kasai et al., 2022) and gamified quality ratings (Scott et al., 2023). Since image accessibility is a fundamental use case of image description generation and evaluation at scale, we adopt the evaluation scheme proposed by Kreiss et al. (2022a). They introduce a set of 5 questions to assess multiple dimensions of description quality, which show a promising correlation between sighted and blind and low vision (BLV) participant judgments.

### 4.1 Stimuli selection

The data was randomly sampled from the English language subset of the WIT dataset (Srinivasan et al., 2021). To provide an in-depth understanding of how model scoring behavior corresponds with human description preferences, we prioritized detailed and high-coverage annotations for each description over increased data sample size. As Sections 4.4 and 5.2 show, the dataset size is sufficient to highlight robust patterns in model behavior.

Our dataset contains 204 sampled data points, each of which consists of an alt text description written by Wikipedia editors as well as the corresponding image and context (article title, first paragraph, section title, section text, caption). Sampling was restricted to data where both an alt description (as it appears in the HTML alt tag) and a caption (visible to everyone below the image) were present (Kreiss et al., 2022b). In WIT's subset of English Wikipedia, 65% of alt descriptions

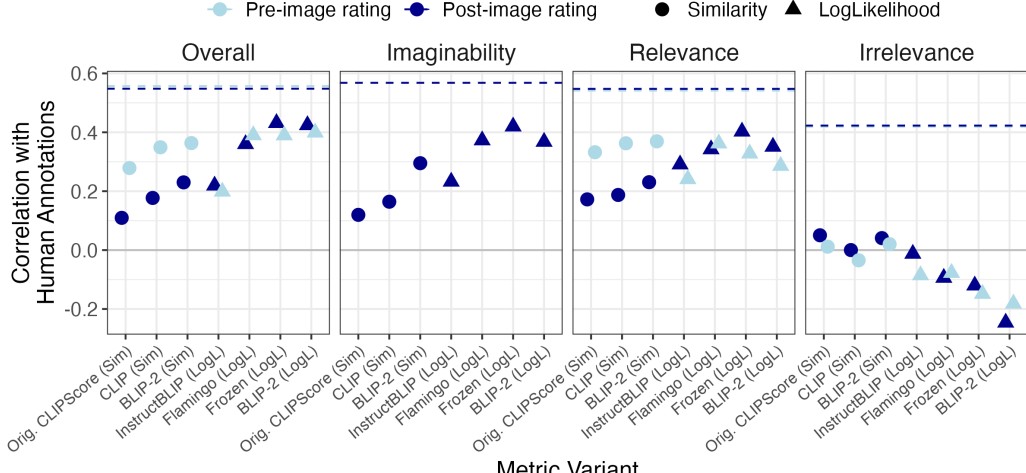

Figure 2: Best correlations with human annotations of each model category for predicting description quality. All correlations for overall, imaginability, and relevance are statistically significant Pearson correlations ($p < 0.001$). No irrelevance correlations are significant. Correlations with ratings participants gave before seeing the image are in light blue, and ratings after seeing the image are in dark blue. The dashed lines represent the estimated correlational ceiling on the data, based on the correlation of the original data with the averaged resampled data. A results table is presented in Appendix E.

are identical to the caption, which is generally discouraged in image accessibility guides (e.g., the WebAIM accessibility guide specifically advises against redundant information[3]). To optimize for most informative sampled data, we therefore subsampled such cases to 20% of the crawled data.

## 4.2 PROCEDURE

Before starting the main study, participants were introduced to the overall goal of making images nonvisually accessible. Then, participants were given 5 images with their associated description that they were asked to rate, which were presented within the available context from the Wikipedia article page. The descriptions were randomly sampled, but each participant saw exactly one description that was identical to the caption and 4 descriptions that were distinct from the caption. Participants rated each description twice, once before and once after seeing the image. After the image was revealed, participants saw what they had previously selected so that they could make an informed decision to either keep or change their rating. Each image was rated based on 6 distinct questions.

Question order was randomized between participants, except that the *overall* quality question always appeared last. Participants were recruited via Prolific (Palan & Schitter, 2018), restricted to US-based workers. The study had a median completion time of 11.5 minutes, and participants received $2.40 compensation ($12.50/hr). We continued recruitment until all descriptions had received at least 3 annotations from workers who passed the attention check (see Appendix A for details on the participant population and data exclusions).

## 4.3 RESULTS: DATASET PROPERTIES

The dataset contains 768 annotations, averaging 3.8 distinct participant ratings for each description (see examples in Appendix Figure A.4). *Overall* ratings are the most intuitive quality measure, which is why they are the focus of the following dataset analyses. Figure 1A shows the distributions of averaged ratings for each of the questions. Specifically, the *overall* ratings show encouraging coverage over the whole scale, which is essential for evaluating the effectiveness of metrics. We also find that quality ratings are significantly correlated with the description length, that descriptions are regarded as less useful when they are identical to the associated caption, and that faulty descriptions consistently receive lower ratings from participants. We include details on these analyses in Appendix B.

## 4.4 RESULTS: CORRELATION WITH REFERENCELESS METRICS

Using the annotated data, we correlate the description quality as predicted by the metrics with the averaged human-annotated description quality. We selected the best-performing model variants based on the highest correlation with the *overall* post-image ratings (see Appendix D for model details).

---

[3] https://webaim.org/techniques/alttext/

Figure 2 shows the Pearson correlations for each model variant with the human annotations for all quality assessment questions. There is a strong qualitative difference in correlation between the ratings participants provided before seeing the image (presented in light blue) vs. after seeing the image (dark blue), specifically for similarity-based metrics (denoted by circles).

Concretely, similarity-based metrics are uniformly less able to capture pre-image quality judgments than post-image ones, which is not borne out for any of the likelihood-based metrics (denoted by triangles). Most strikingly, this pattern even holds within the same model type (BLIP-2), suggesting that the scoring method itself introduces a robust semantic bias for evaluating descriptions. These differences trace mainly to the descriptions marked as containing inaccurate information (see Appendix F).

While all similarity-based metrics are less successful in predicting pre-image ratings, we place more emphasis on the post-image ratings for two reasons. First, when establishing the annotation scheme, Kreiss et al. (2022a) note that sighted participant ratings after seeing the image show slightly higher correlation with blind and low vision participant judgments. Second, it is only after seeing the image that sighted users can evaluate whether descriptions are truthful. In the post-image condition, most metrics achieve reasonably high correlations with the human ratings (with $r \approx 0.4$), except for InstructBLIP ($r = 0.2$). Nevertheless, the distinction in correlation with the pre-image ratings already points to a qualitative difference between likelihood- and similarity-based metrics and the role that image–text alignment plays for achieving this correlation. This is further supported by high correlations of the predicted ratings within those categories, but not across (see Appendix C).

Based on the correlation with human ratings, these results seem to tell a promising story for the potential of leveraging powerful pretrained models out-of-the-box for referenceless image description evaluation. The by-question and across-metric correlational analyses, however, indicate qualitative differences in the way that the metrics assign these scores.

## 5 CONTEXTREF: EVALUATING ROBUSTNESS

While the high correlations of the metrics with human ratings are reassuring, they provide only limited insight into how the metrics work and where they fail. Based on prior work on what makes descriptions (not) useful and the type of errors language and vision models often make, the second part of ContextRef introduces dataset augmentations which any metric should be expected to be sensitive to. These augmentations are in contrast to many previous approaches testing whether models are **in**sensitive to perturbations (e.g., Qiu et al. 2022; Rohrbach et al. 2018). Here, we expect all augmentations to necessarily result in lower scores than are assigned to the ground-truth data.

### 5.1 DATA AUGMENTATIONS

The applied data augmentations manipulate a subset of three potential causes of errors: missing image–text alignment, over-reliance on string predictability, and lack of contextual sensitivity. We exemplify each augmentation in Figure 1B.

**Shuffled descriptions** Descriptions are shuffled to be assigned to a different image from the dataset. This tests whether a metric integrates image and description information jointly and is commonly used to uncover object hallucinations (Radford et al., 2021; Hessel et al., 2021; Cui et al., 2018).

**Shuffled contexts** Contexts that each image originated from are shuffled. Prior work found that if the connection between the image and the context it appears in isn't apparent from the description, it receives low quality ratings, especially from BLV participants (Kreiss et al., 2022a).

**Shuffled words** Prior work suggests that grammaticality is an indicator of description quality (Kasai et al., 2022; Mitchell et al., 2012; Elliott & Keller, 2013). Shuffling word order is a long-standing strategy to investigate sensitivity to grammaticality (Barzilay & Lee, 2004; Cao et al., 2020; Parthasarathi et al., 2021) and some Transformer-based language model variants can be trained to effectively perform language modeling without consideration to word order information (Sinha et al., 2021; Abdou et al., 2022). In addition to string predictability, word shuffling can also affect image–text alignment since, for instance, property attribution can become ambiguous (e.g., "a red shirt and blue pants" can become "blue shirt pants a red and").

**Proper name replacement** We used GPT-4 (OpenAI, 2023) to identify and replace all proper names in the descriptions, such as people's names or locations, with likely alternatives.[4] The accuracy of

---

[4]Using GPT-4 allowed for more naturalistic replacements than could be done with pattern-based methods.

proper nouns based on the image alone is generally difficult to verify but essential for error detection. Following the same logic, we also replaced dates in this manipulation. 104 out of the 204 descriptions contain at least one proper name replacement.

**Frequent alignment errors** Previous work has established a number of common errors that image description generation models make, including the misidentification of colors, clothing items, or people's ages (van Miltenburg & Elliott, 2017). We used GPT-4 to detect and replace those terms with incongruent alternatives in order to necessarily make the description inaccurate. 153 out of the 204 descriptions contain at least one induced common model error.

**Frankenstein images** A random object (e.g., a golden crown) is saliently placed within the image at a random position (Yu et al., 2022a). The score for a description that doesn't mention the added object is expected to be lower due to the salience of the image manipulation. This tests image–text alignment but would likely also be reflected in metrics sensitive to image coherence.

**GPT-2 continuations (long/short)** To test the effect of string predictability on the predicted rating (Rohrbach et al., 2018), descriptions were extended by an additional sentence (*long* condition). We used GPT-2 (Radford et al., 2019) to generate likely string continuations that are not grounded in the image. To account for the length artifact, we also created a version where GPT-2 completes the first half of the description (*short* condition). This tests image–text alignment by adding image-independent information that is highly likely.

**Irrelevant final sentence** To further exaggerate the condition of adding irrelevant but high-probability strings, we add an irrelevant sentence to the end of a description. The sentence is randomly chosen from 10 sentences from Wikipedia, e.g., "The elephant is the largest existing land animal."

**Exact repetition** Inspired by the observation that language models tend to repeat phrases (Holtzman et al., 2019; Xu et al., 2022; Tang et al., 2023), we add a test for an exact repetition of the description. Reference-based evaluation metrics can show a bias towards long sentences with repeated phrases (SPICE; Liu et al. 2017). Redundant information should be dispreferred by a metric for two reasons. First, redundant information can lead to undesired pragmatic inferences (Nie et al., 2020), and second, accessibility technologies like screen readers make it hard to skip ahead and avoid redundant parts.

## 5.2 RESULTS

To contextualize the behavior of the various metrics for each augmentation type, Figure 3 shows the exact number of descriptions for which the metrics assigned the same, lower, or higher scores. Given the nature of the augmentations, a well-calibrated metric should assign a lower score for all augmented descriptions, resulting in all green bars. Cases where the metrics are insensitive to the augmentation are marked in light pink. The most problematic cases are marked in dark pink. Here, the metric considers the augmented data to be of higher quality than the ground truth.

No metric passes all data augmentations out-of-the-box. Across augmentation variants, augmented descriptions often counter-intuitively receive a higher score than their ground-truth counterparts (see Appendix G for a complementary analysis of the average assigned scores). This illustrates fundamental shortcomings of simply selecting referenceless metrics based on human correlation performance alone, and shows how those metrics can mislead model development based on their behavior on likely model error patterns.

The data augmentation results further support the previous observation that similarity-based and likelihood-based metrics show distinct semantic sensitivities. Notably, they strongly differ in their sensitivity to *shuffled descriptions*. CLIP correctly decreases the score for almost all shuffled descriptions, providing evidence that the task is well-defined. The original CLIPScore and BLIP-2 are similarly successful, which is perhaps unsurprising given the contrastive learning objective underlying the scores and provides further evidence that similarity-based metrics are sensitive to image–text mismatches. However, the Frozen metric, which showed a comparatively strong correlation with the human data, increases its score for more than 25% of all incompatible descriptions, and the best-performing BLIP-2 does so for more than half. This pattern is similarly reflected in the *Frankenstein images* augmentation and suggests a key failure case of the likelihood-based metrics.

When it comes to *shuffled contexts*, however, likelihood-based metrics appear comparatively more successful. Even the previously proposed contextual CLIPScore variant that showed encouraging correlations with sighted and BLV user rating (Kreiss et al., 2022a) fails when the contexts are

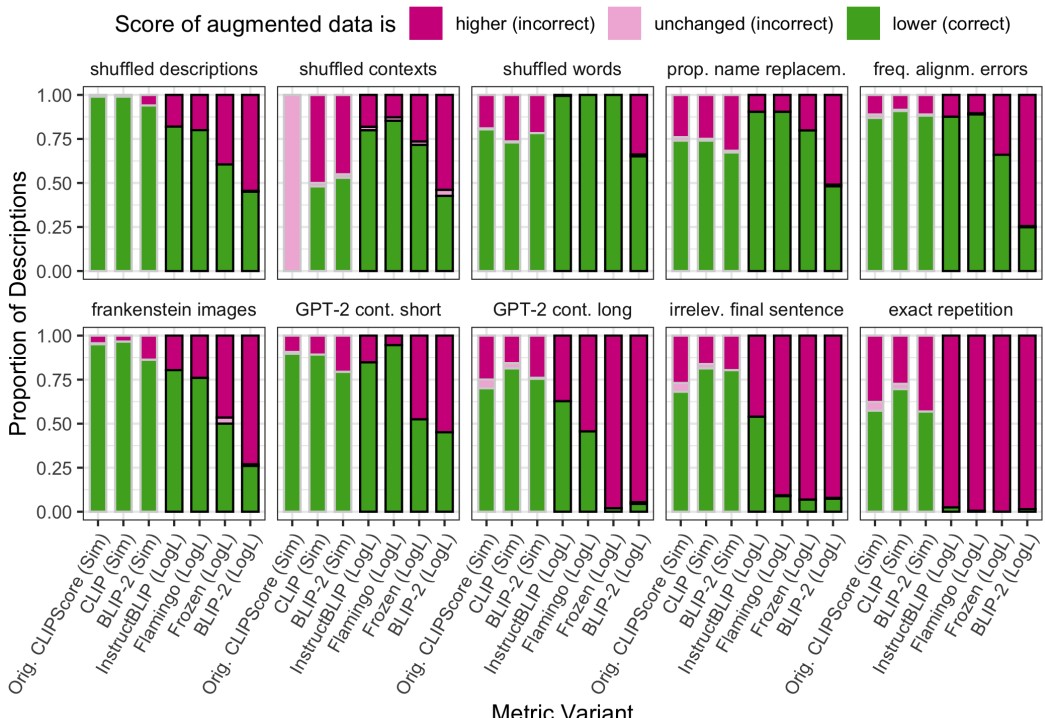

Figure 3: Proportion of augmented descriptions that receive lower scores (green), unchanged scores (light pink), or counter-intuitively higher scores (dark pink). Metrics are sorted according to their correlational performance with the human judgments in Figure 2. Across augmentations, models commonly assign higher scores to augmented descriptions that by definition contain wrong or irrelevant irrelevant, omit relevant information, or are ungrammatical.

randomly shuffled. Another success story for likelihood-based scores is the *shuffled words*, where they achieve ceiling accuracy. In 25% of the descriptions, the similarity-based metrics CLIP and BLIP-2, however, assign a higher score to the shuffled descriptions than their ordered counterparts.

The most striking failure case of likelihood-based metrics is the strong preference for descriptions that were augmented to increase the predictability of the string (*GPT-2 continuation long, irrelevant final sentence*, and *exact repetition*). For *exact repetition*, all likelihood-based metrics show a categorical preference for the augmented description over the original one, which is only marginally improved for the case where a correct but completely *irrelevant final sentence* is added. This suggests that increased string predictability (independent of the image) biases especially likelihood-based metrics towards higher scores. This is in line with the prior observation that language models trained for description generation exhibit strong language priors (Rohrbach et al., 2018).

In sum, all models exhibit unexpected behavior and assign higher scores to descriptions that are decidedly worse. However, similarity- and likelihood-based metrics show distinct sensitivity patterns across augmentations. Likelihood-based metrics are highly influenced by added irrelevant information and show a comparatively low sensitivity for detecting descriptions that don't belong to an image. However, they are very sensitive to manipulations of word order and context. Interestingly, Instruct-BLIP had the lowest correlation with human ratings but seems more sensitive to data manipulations than the on-the-surface more promising likelihood-based alternatives.

Based on the behavior on augmented data, similarity-based metrics appear more promising since they consistently judge at least half of all augmented descriptions as worse compared to their original counterpart. However, increased scores for the augmented examples are still present at an alarming rate, and the similarity-based metrics seem to fail to respond meaningfully to context perturbations.

## 6  TOWARDS BETTER METRICS VIA FINE-TUNING WITH CONTEXTREF

While out-of-the-box referenceless metrics appear promising in terms of correlation with human judgments, they exhibit a wide range of unexpected behaviors on data augmentations that target

image–text alignment, predictability of the string, and context sensitivity. In this section, we explore the extent to which fine-tuning can guide metrics toward capturing the reduced quality associated with these expected model-made errors in the augmentations. This is partially motivated by recent trends in research on large pretrained models, where techniques such as instruction tuning and reinforcement learning from human feedback (RLHF) have been used to help close gaps between the capabilities of large pretrained models and real-world usage (Ziegler et al., 2019).

We select CLIP, a similarity-based metric that is the most robust against the data augmentations, and Frozen, a likelihood-based metric that had particularly strong overall correlation with human ratings but still some promising scoring behavior on the data augmentations. We split the data into an 80% train and 20% test split, ensuring that any augmentations involving data shuffling are only shuffled within the respective split to avoid contamination of the test set.

We first trained the best-performing CLIP model for 0.5 epochs with a learning rate of $5e^{-6}$ and a batch size of 64, with the Adam optimizer (Kingma & Ba, 2014). Fine-tuning CLIP solely on the data augmentations results in deterioration of the human judgment correlation. When reaching 0.5 epochs, CLIP achieves some performance improvements in 7 out of 10 augmentations but only at the cost of reducing the Pearson correlation with the human judgments from 0.36 to 0.27.

| Dataset variant | CLIP | | Frozen | |
| --- | --- | --- | --- | --- |
| | Untuned | Tuned | Untuned | Tuned |
| shuffled descr. | 100.0 | 100.0 | 66.7 | **69.2** |
| shuffled contexts | 43.9 | **48.8** | 58.5 | **65.9** |
| shuffled words | 67.6 | **91.9** | 100.0 | 100.0 |
| proper name repl. | 76.2 | **81.0** | 85.7 | 85.7 |
| freq. align. errs. | 89.3 | 89.3 | 71.4 | **75.0** |
| frankenstein img. | 100.0 | 100.0 | 53.7 | 53.7 |
| GPT-2 cont. short | 78.1 | **90.2** | 61.0 | **63.4** |
| GPT-2 cont. long | 65.9 | **100.0** | 2.4 | **9.8** |
| irrel. final sent. | 80.5 | **100.0** | 2.4 | **19.5** |
| exact repetition | 65.9 | **100.0** | 0.0 | 0.0 |

Table 1: Model performance (percent) on dataset augmentations before and after jointly fine-tuning on the augmentations and human judgments. Accuracy is the proportion of descriptions in the test set that receive the expected lower score compared to the ground-truth.

To mitigate this issue, we jointly trained on the augmented data and the raw evaluation scores from the human-subjects experiment (Section 4). For this training, we maintain other hyperparameters, but change the learning rate to $2e^{-6}$. While still maximizing for the Pearson correlation with human judgments on *overall* (post-image) ratings (from 0.36 to 0.30), fine-tuned CLIP achieves remarkable performance gains on the data augmentations, shown in Table 1. Augmentations with the highest gains are *shuffled words* ($+24\%$), and perfect performance on *GPT-2 continuation long* ($+34\%$), *irrelevant final sentence* ($+20\%$), and *exact repetition* ($+24\%$). For the *shuffled contexts* augmentation, fine-tuned CLIP also improves performance, but doesn't change its score in 9% of the descriptions and provides a higher score for about 40% of the augmented data compared to the ground truth.

Fine-tuning Frozen jointly on the human data and data augmentations also improves performance on many of the data augmentations, but it still largely falls behind CLIP. Even with fine-tuning, Frozen can't get any traction on *exact repetition* and still largely provides higher scores for descriptions containing irrelevant information (*GPT-2 continuation long* and *irrelevant final sentence*).

These fine-tuning results highlight how fine-tuning existing models to align with common model shortcomings can be an effective strategy for developing more intuitive referenceless metrics. For CLIP, a similarity-based metric, fine-tuning can alleviate most of the unintuitive behavior. However, context-sensitivity remains challenging, suggesting that especially a successful integration of context might require more fundamental innovations to successfully guide metric alignment with people's judgments.

## 7 CONCLUSION

Referenceless image description evaluation metrics can support and promote fast progress on image description generation models, but only if they reliably correlate with human preferences. We introduce ContextRef, a benchmark for assessing these metrics against the results of a human-subjects experiment and against data augmentations that should systematically make descriptions worse. We find that no metric excels across all parts of ContextRef, but careful fine-tuning improves metric performance. Integrating context remains a challenge, though; we hope that ContextRef spurs new research on this important aspect of image description generation.

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

## A    RECRUITMENT DETAILS AND DATA EXCLUSIONS

**Participants** Participants were recruited on the online study platform Prolific. Recruitment was restricted to US-based participants using Prolific's standard sampling. Participant age ranged from 19 to 84 years (mean: 37.9; median: 36; 3 participants didn't report their age). 96% of participants indicated that English is among their native languages (4 participants chose not to answer). We didn't collect gender information since collecting a variety of personal information can lead to compromising participant identity, and gender isn't central to our study setup. The study ran under an IRB protocol. In Appendix I, we discuss how our participant pool and individual variation pose important limitations for the applicability of this work to immediate downstream application.

**Exclusions** Participants were excluded based on their response to the *Added info* question in the case where the caption and description are identical. The question asks how much information the description adds that's not present elsewhere. We excluded participants who gave a rating of 3 or higher either before or after seeing the image. Based on this attention check, we excluded the work of 190 out of 358 unique participants (53%), which is in line with recent estimates on the proportion of low-quality work on Prolific (Douglas et al., 2023).

## B   ADDITIONAL DATASET PROPERTIES

The *overall* quality ratings are significantly correlated with the length of the descriptions ($r = 0.27$, $p < 0.001$), which is at odds with previous work that only found length correlations for blind and low vision participants (Kreiss et al., 2022a). The conflicting results could be due to the high statistical power in our dataset (with 768 as opposed to about 200 annotations).

The *overall* quality ratings confirm the assumption made in prior work and accessibility guides that descriptions are considered less useful when they are an exact duplicate of the caption (Welch two sample t-test: $t = -6.24$, $df = 279.01$, $p < 0.001$). Our results provide empirical evidence to the normative arguments that alt descriptions should complement the textually available information instead of repeating it, and emphasizes the importance of considering the textual context the image is in for naturalistic description evaluation.

For 25 images (12% of all images) at least one annotator reported that the description contains potentially wrong information. For 6 of those images (3% of all images), more than half of the annotators reported potentially wrong information. Participants note potential mistakes in the image-text alignment ("The aircraft can be seen in the bottom left, not the bottom right of the image.", or "This is not a statue"), misspellings ("'Photo' is spelled incorrectly"), and contextual misalignment of the image/description and the rest of the article ("The image isn't relevant to the article; it shows Cuba and the surrounding area."). Images with potentially faulty descriptions are rated lower on average in the post-image but not pre-image condition (Welch two sample t-test: $t = -5.48$, $df = 135.37$, $p < 0.001$), suggesting that most of those judgments are based on image-text alignment issues that can only be verified based on the image.

## C   CROSS-METRIC CORRELATIONS

Figure A.5 shows the correlations between all metrics, and supports a clustering of metrics based on how scores were obtained. Similarity-based metrics and likelihood-based metrics correlate highly amongst each other respectively ($r = [0.33, 0.87]$), but show much less correlation across ($r = [0.05, 0.42]$, marked in yellow).

## D   BEST PERFORMING MODEL SPECIFICATIONS

We selected the models that had the highest correlation with the human judgments based on the overall description quality ratings participants provided after the image was revealed. Those were the following model variants which are presented and further analyzed in the paper.

### D.1   LIKELIHOOD-BASED METRICS

**Flamingo** For our Flamingo model, we use OpenFlamingo v2's 9 billion parameter model, combining a 7 billion parameter text model (MosaicML's MPT-7B) and CLIP ViT L/14 (large with a patch size of 14 pixels) encoder using cross-attention (Awadalla et al., 2023).

**Frozen** For our Frozen evaluations, we use GPT-2 large as the language model (700 million parameters) combined with the EVA-02 image encoder model based on CLIP (Fang et al., 2023). In particular, we use the "enormous" EVA model size (4.4 billion parameters) with a patch size of 14 pixels, which was pretrained on the LAION2b dataset (Schuhmann et al., 2022), through OpenCLIP (Ilharco et al., 2021).

**BLIP** For BLIP and InstructBLIP, we use the corresponding BLIP-2 variants with Flan-T5 XXL as a base model (which has 11 billion parameters) (Li et al., 2023).

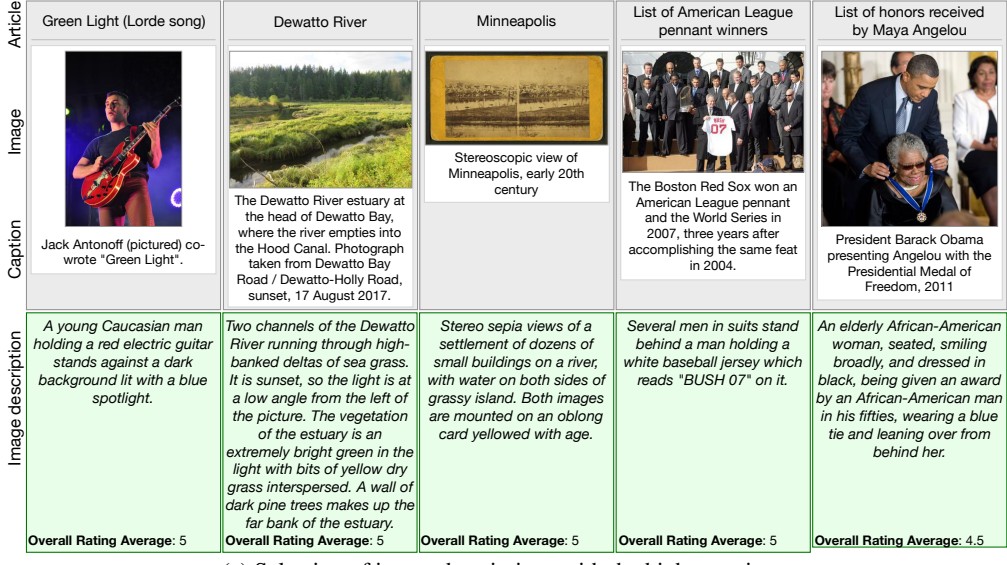

(a) Selection of image descriptions with the highest ratings.

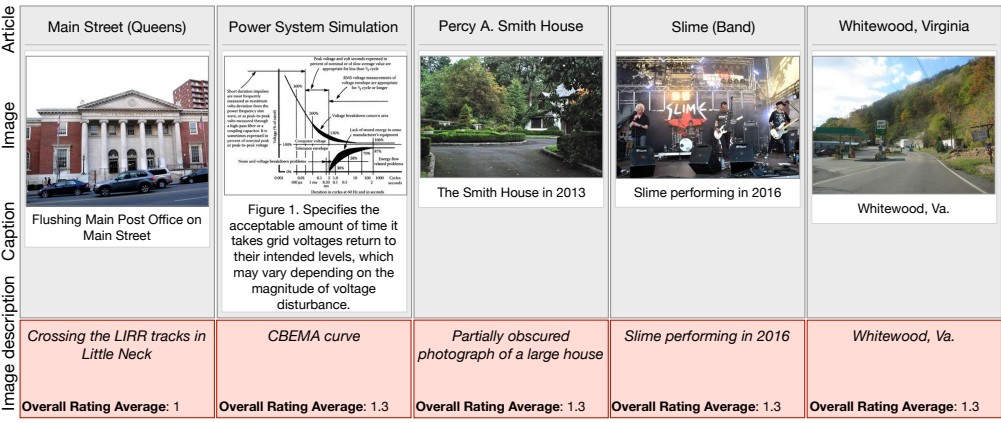

(b) Selection of image descriptions with the lowest ratings.

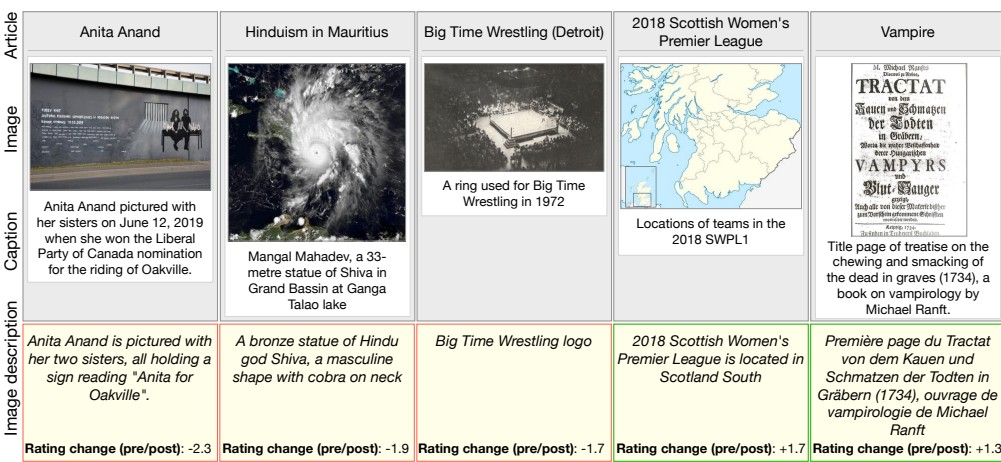

(c) Image descriptions with the highest change average between ratings before participants saw the image and after. In the first three cases, images seem wrongfully placed and ratings decreased when the image was revealed. For the last two cases, ratings increased after the image was revealed.

Figure A.4: Examples from the human-annotated dataset.

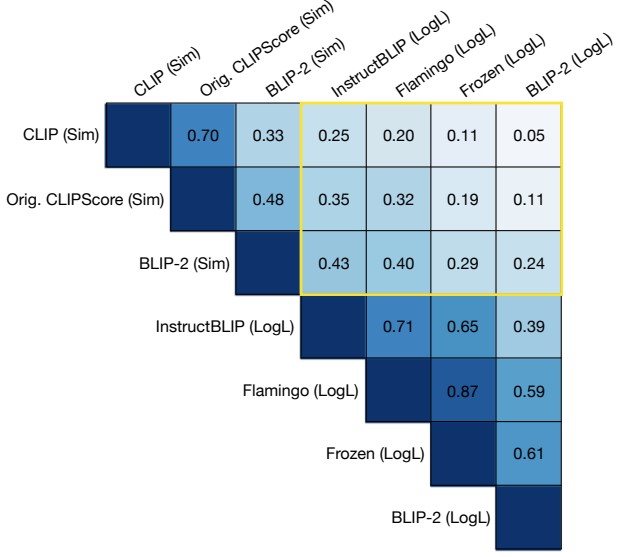

Figure A.5: Correlations between metrics ordered by the largest eigenvalues of the correlation matrix. Correlations within similarity-based and likelihood-based metrics respectively are higher than correlations between these categories (marked in yellow).

## D.2 SIMILARITY-BASED METRICS

**Original CLIPScore** For the original CLIPScore, we use CLIP ViT B/32 (base with a patch size of 32 pixels) as is done in their original paper (Hessel et al., 2021).

**CLIP** For the selected contextual CLIP-based metric, we use a CLIP-based ViT B/16 (base with a patch size of 32 pixels), trained on the LAION-400M dataset for 32 epochs, from OpenCLIP (Ilharco et al., 2021).

**BLIP** For BLIP, we use the BLIP-2 'feature extractor' for our similarity-based metrics, based on CLIP ViT-G (giant with a patch size of 14 pixels) (Li et al., 2023).

## E DETAILED CORRELATION RESULTS

Table A.1 lists the Pearson correlation values presented in Figure 2.

## F QUALITATIVE ANALYSIS: PRE- VS. POST-IMAGE CORRELATIONS

A qualitative follow-up analysis reveals that the difference in pre- vs. post-image rating correlations is primarily due to the descriptions that participants indicated to contain potentially wrong information. In other words, because the metrics always have access to the images, they penalize incorrect descriptions, while participants cannot know descriptions are incorrect before they've seen the images. Excluding those descriptions, the similarity-based metric correlations rise close to post-image correlation rates. CLIPScore reduces the correlation difference from 0.17 to 0.03, CLIP reduces it from 0.17 to 0.09, and BLIP-2 reduces it from 0.13 to 0.05. These results indicate that much of the difference in pre- and post-image correlation is driven by image–text incongruencies sighted users assess only when viewing the image.

## G DATA AUGMENTATIONS: AVERAGE SCORE ANALYSIS

Figure A.6 shows the average description score for each augmented dataset assigned by the different models. The original dataset is presented in black and is the average score that all data augmentations are compared against. Given the nature of the augmentations, a well-calibrated metric should assign a lower average score throughout. The cases where this is successful are marked in green. If the

| Question | Metric Variant | Rating |
|---|---|---|
| Overall | Orig. CLIPScore (Sim) | pre-image: 0.11; post-image: 0.28 |
| Overall | CLIP (Sim) | pre-image: 0.18; post-image: 0.35 |
| Overall | BLIP-2 (Sim) | pre-image: 0.23; post-image: 0.36 |
| Overall | InstructBLIP (LogL) | pre-image: 0.22; post-image: 0.20 |
| Overall | Flamingo (LogL) | pre-image: 0.36; post-image: 0.39 |
| Overall | Frozen (LogL) | pre-image: 0.43; post-image: 0.39 |
| Overall | BLIP-2 (LogL) | pre-image: 0.42; post-image: 0.40 |
| Imaginability | Orig. CLIPScore (Sim) | pre-image: 0.12 |
| Imaginability | CLIP (Sim) | pre-image: 0.16 |
| Imaginability | BLIP-2 (Sim) | pre-image: 0.30 |
| Imaginability | InstructBLIP (LogL) | pre-image: 0.23 |
| Imaginability | Flamingo (LogL) | pre-image: 0.37 |
| Imaginability | Frozen (LogL) | pre-image: 0.42 |
| Imaginability | BLIP-2 (LogL) | pre-image: 0.37 |
| Relevance | Orig. CLIPScore (Sim) | pre-image: 0.17; post-image: 0.33 |
| Relevance | CLIP (Sim) | pre-image: 0.19; post-image: 0.36 |
| Relevance | BLIP-2 (Sim) | pre-image: 0.23; post-image: 0.37 |
| Relevance | InstructBLIP (LogL) | pre-image: 0.29; post-image: 0.24 |
| Relevance | Flamingo (LogL) | pre-image: 0.34; post-image: 0.36 |
| Relevance | Frozen (LogL) | pre-image: 0.40; post-image: 0.33 |
| Relevance | BLIP-2 (LogL) | pre-image: 0.35; post-image: 0.29 |
| Irrelevance | Orig. CLIPScore (Sim) | pre-image: 0.05; post-image: 0.01 |
| Irrelevance | CLIP (Sim) | pre-image: 0.00; post-image: -0.03 |
| Irrelevance | BLIP-2 (Sim) | pre-image: 0.04; post-image: 0.02 |
| Irrelevance | InstructBLIP (LogL) | pre-image: -0.01; post-image: -0.09 |
| Irrelevance | Flamingo (LogL) | pre-image: -0.09; post-image: -0.08 |
| Irrelevance | Frozen (LogL) | pre-image: -0.12; post-image: -0.15 |
| Irrelevance | BLIP-2 (LogL) | pre-image: -0.25; post-image: -0.18 |

Table A.1: Pearson correlations of metric predictions with human judgments. This supplements Figure 2.

augmentation has no or barely any effect on the average score (with overlapping confidence intervals), it is marked in light pink. In these cases, the metrics appear insensitive to the manipulation. The most problematic cases are marked in dark pink. Here, the metric considers the augmented data to be of higher quality than the ground truth. We now turn to the central qualitative observations. Metrics are sorted according to their correlational performance with the human judgments in Figure 2.

No metric passes all data augmentations. The degree of unexpected behavior increases with models that show higher out-of-the-box correlations with human ratings. While CLIP scores in 8 out of 10 augmentations on average penalize the augmented data, BLIP-2 never assigns lower average scores in any data augmentation.

For similarity-based metrics, the *shuffled descriptions* augmentation stands out, as the average assigned scores are much lower than for any other augmentation. This is perhaps unsurprising given the contrastive learning objective underlying the scores and further evidence that similarity-based metrics are sensitive to image-text mismatches.

Surprisingly, all likelihood-based metrics assign a higher average score in at least one data augmentation. Specifically, they all tend to fail when additional information is added that's either unrelated to the image (*GPT-2 continuation long* and *irrelevant final sentence*) or an exact repetition of the description (*exact repetition*). Taken together with the strict dispreference for descriptions where the words are shuffled (*shuffled words*), this suggests that the language model plays an elevated role over the vision model for the likelihood-based metric scores. This is in line with the prior observation that language models trained for description generation exhibit strong language priors (Rohrbach et al., 2018). In further support of this hypothesis is the fact that these metrics are fairly insensitive to the *Frankenstein images* augmentation that introduces an image-internal incongruency that should lead to worse image-text alignment since there is insufficient detail. Note, however, that this insensitivity

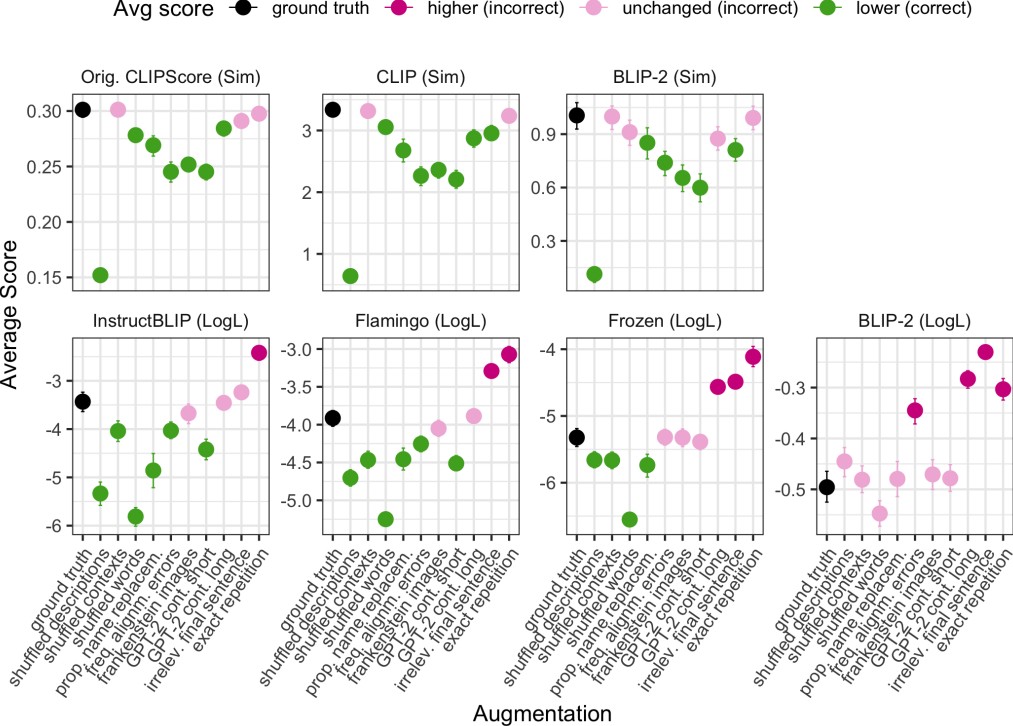

Figure A.6: Most correlated metrics' average score for each data augmentation. Data augmentations resulting in a lower average score are presented in green, unchanged scores are presented in light pink and data augmentations counter-intuitively resulting in better scores are presented in dark pink. Error bars indicate 95% bootstrapped confidence intervals computed over all assigned scores.

is only borne out for image-independent or redundant information. Descriptions containing false information are generally dispreferred (see *shuffled descriptions* and *proper name replacement*), suggesting at least some image-text coordination.

Remarkably, the increase in average score in the likelihood-based models is often quite large compared to the lower scores assigned in other augmentations. In Frozen, while shuffling the descriptions only results in a minor decrease in average scores (*shuffled descriptions*), simply repeating the description twice results, on average, in a 3 times higher score. This suggests there is significant variation in sensitivity to different data augmentations.

## H  MODEL DETAILS

### H.1  LIKELIHOOD-BASED PROMPT

We use the following prompts for evaluating the likelihood of text with the language models. The default arguments used are included, where 'reduce' controls whether to average the likelihood over the tokens, and 'diff' whether to calculate the log-likelihood of just the target. The score functions are used to evaluate the likelihood of the text. We ultimately use text_if_good with reduce but without diff.

```
1 def text_if_good(self, image_name, text, context, reduce=True, diff=
    ↪ False, return_tensor=False):
2     base_text = ""
3     if context is not None:
4         base_text += f'[Context: {context}] '
5     base_text += 'High quality, accessible, image description:'
6     target_text = text
```

```
 7       score = self.score_fn(image_name, base_text, target_text, reduce=
   ↪ reduce, diff=diff, return_tensor=return_tensor)
 8       return score
 9
10 def good_if_text(self, image_name, text, context, reduce=True, diff=
   ↪ False, return_tensor=False):
11     base_text = ""
12     if context is not None:
13         base_text += f'[Context: {context}] Look at the context, photo,
   ↪  and description and rate the description from 1-5 based on
   ↪ whether it is a high quality, accessible, image description.
   ↪ Description:'
14     else:
15         base_text += 'Look at the photo and description and rate the
   ↪ description from 1-5 based on whether it is a high quality,
   ↪ accessible, image description. Description:'
16     target_text = '5'
17     score = self.score_fn(image_name, base_text, target_text, reduce=
   ↪ reduce, diff=diff)
18     return score
```

## H.2    FROZEN INTUITION

For the Frozen baseline, we match the tokens in the language model's embeddings to their corresponding words in the multi-modal model's embedding space, and create a new token corresponding to the linear combination of the tokens in the multimodal space for a new image. For example, consider an image of a "pluot" that is represented in the multimodal model's embedding space as a linear combination of its embeddings for the words plum and apricot: i.e., $\text{encode\_image}(pluot\_image) = \alpha * \text{encode\_text}(plum) + \beta * \text{encode\_text}(apricot)$. Then, a new token would be created in the language model's vocabulary corresponding to the same linear combination of the language model embeddings for plum and apricot: $\text{new\_token}(pluot\_image) = \alpha * \text{embed\_token}(plum) + \beta * \text{embed\_token}(apricot)$. Then, the image can be passed into the language model as if it is a token.

## I    LIMITATIONS

The aim of ContextRefis to test metric alignment with people's intuitions on accessibility description quality. To get initial traction on this task, we largely abstract away from individual participant variation and focus on the average ratings across participants. We do that by obtaining multiple ratings for each individual description which reduces the effects of outliers and instead reflects general preference patterns in the recruited population. Our approach therefore has two fundamental constraints.

Firstly, our empirical data might reflect population-level biases based on shared experiences in our participant population. Our participant population is US-based, and due to the recruitment medium primarily consists of tech-savvy, young, and non-blind adults. Prior work has shown that depending on for example the cultural background, referential preferences change, leading to model behavior such as only accurately recognizing *Western* wedding dresses (Shankar et al., 2017). These referential choices are important for building appropriate and equitable systems, but don't necessarily fall out this framework.

Secondly, we largely abstract away from the individual variation within our dataset. Individual preference variation is an important component of current image description research and is still underexplored. One such example is description length. A variety of papers have found conflicting evidence on whether shorter or longer descriptions are generally preferred (Slatin & Rush, 2003; Petrie et al., 2005; Rodríguez Vázquez et al., 2014). We therefore chose to set up this study based on a goal-oriented framing (consider the purpose and choose your responses accordingly) as opposed to explicit training and instructions. We thereby focus on investigating communicatively motivated intuitions, instead of alignment to prescriptive practices. However, we encourage work that builds on ContextRefto adjust the experimental design according to potentially distinct downstream needs, and the framework easily extends to instruction-centric experiment design.

## J    EXTENDED RELATED WORKS

### J.1    REFERENCELESS METRICS

Metrics for evaluating image description quality can be reference-based or referenceless. Reference-based metrics originated in Machine Translation and assign a quality score based on the correspondence of a proposed description to multiple ground-truth references (Papineni et al., 2002). These metrics tend to be image- and context-independent since the ground-truth references are assumed to capture all relevant information. However, these metrics often require that there are multiple high-quality ground-truth references to be useful (Anderson et al., 2016), which severely limits their application.

Recently, a number of referenceless evaluation metrics have been proposed, which assign a quality score for a description without requiring ground-truth references. Instead, text quality is evaluated by considering either its connection to the image content (Hessel et al., 2021; Lee et al., 2021a;b; Scott et al., 2023) or its quality as standalone text (Feinglass & Yang, 2021). CLIPScore (Hessel et al., 2021) and UMIC (Lee et al., 2021b) define it as a classification problem where a model is trained contrastively to distinguish compatible and incompatible image–text pairs. QACE (Lee et al., 2021a) returns a high score when a model returns similar answers to questions based on the image and text. The metric proposed by Scott et al. (2023) is based on training a model on quality ratings that people assigned to image–text pairs and is grounded in a gamified experimental design setup that reduces interannotator variation. Finally, SPURTS is the referenceless subpart of SMURF (Feinglass & Yang, 2021), and the only referenceless metric listed here, which is image-independent.

In this paper, we explore the general benefit of large pretrained Vision-Language models for the task of referenceless image description evaluation. We investigate a wide range of models and ways to obtain a quality score. We further introduce a pipeline that highlights the limitation of simply relying on correlations with human ratings and uncovers fundamental shortcomings for current state-of-the-art models when used without fine-tuning. Our work proposes a benchmark and pipeline to help referenceless metric development and assessment, which is (in contrast to all prior efforts) grounded in the context where the image appears.

### J.2    IMAGE DESCRIPTION EVALUATION

Even just defining what makes an image description good and useful is challenging. Bernardi et al. (2016) provide an overview of the various dimensions prior research explored for determining quality, including the accuracy of a description (Li et al., 2011; Mitchell et al., 2012; Kuznetsova et al., 2012; Elliott & Keller, 2013; Hodosh & Hockenmaier, 2013; Yatskar et al., 2014; Jiang et al., 2019), whether it's grammatical (Yang et al., 2011; Mitchell et al., 2012; Kuznetsova et al., 2012; Elliott & Keller, 2013), creative (Li et al., 2011), or human-like (Mitchell et al., 2012). More recently, Kasai et al. (2022) introduced THumB, a rubric-based human evaluation framework for image-based text generation models. Participants rate image-based texts along two dimensions in a tradeoff between the accuracy of the information and whether all salient content is mentioned.

In this work, we build on the annotation pipeline introduced in Kreiss et al. (2022a), which has a rubric-based annotation design specifically focused on capturing the dimensions relevant for nonvisual accessibility (e.g., whether all relevant information are mentioned). Based on prior work specifically on model-generated errors (Kasai et al., 2022; Elliott & Keller, 2013), we supplement this annotation by introducing data augmentations that specifically pick up on common model-made mistakes, like truthfulness or ungrammaticality.

