# OpenReview forum: "ContextRef: Evaluating Referenceless Metrics for Image Description Generation"
_ICLR.cc/2024/Conference — ICLR 2024 poster_

### Official Review · Reviewer_T1dS · 2023-10-31

**Soundness:** 3 good
**Presentation:** 3 good
**Contribution:** 2 fair
**Rating:** 6
**Confidence:** 3

**Summary:**

This paper presents ContextRef, a new benchmark designed to assess the alignment of referenceless image description quality metrics with human preferences. The proposed benchmark incorporates human-subject ratings across diverse quality dimensions and ten robustness checks to gauge metric performance under varying conditions. Additionally, the study delves into several metric approaches, revealing that none of them prove successful on ContextRef due to their insensitivity to fundamental changes in examples. Lastly, the paper suggests that careful fine-tuning has the potential to enhance metric performance while acknowledging the ongoing challenge of effectively accounting for context.

**Strengths:**

1. This paper is well-written and easy to understand.
2. The introduction of the new benchmark, ContextRef, is indeed a valuable contribution to the field. A crucial feature of ContextRef is the presentation of images and descriptions within a contextual framework, which plays a crucial role in generating appropriate descriptions.

**Weaknesses:**

While the paper highlights the underperformance of existing methods when dealing with context, this finding is unsurprising to me since the tested models were not trained with context.

**Questions:**

Please refer to the weakness part.

---

> ### Author Response · Authors · 2023-11-17
>
> Thank you for the encouraging comments and supportive feedback! Indeed, the relatively poor performance of these models in scoring image descriptions in context did not come as a significant surprise to us – prior work showing the poor performance of referenceless metrics in this setting (e.g. [1]) was a significant motivation for this work. Our paper establishes a solid empirical basis for this claim, offers new evidence that context is crucial, and points the way to variants of these metrics that are more successful, with new data and new fine-tuning results.
>
> At the same time, one may find the poor performance of likelihood-based models somewhat surprising because, indeed, one may argue that these models are specifically trained with context. In particular, language models have been consistently shown to extract remarkably much from information simply presented in their context windows; this property of large pretrained language models has been the subject of extensive interest in NLP [2,3].
>
> Thus, we’d like to emphasize that, 1) while the observation that these models perform poorly in context is in line with prior work, the universality of this property was likely not something we could have anticipated a priori and 2) we expect there may be additional strategies that leverage pretrained models in novel ways to better incorporate contextual information into existing referenceless metrics, as done in the contextual CLIPScore from [1] – in fact, this is one of the motivations for creating this benchmark.
>
> Finally, independent of how surprising the poor performance of models for integrating context might be, our goal is that this framework will help to push the development of metrics that are sensitive to image context and therefore lead to improvements in the theoretical approach to the problem and, eventually, the downstream usefulness of those systems.
>
> Based on your comment, we further highlighted the context integration procedures in Section 3 and Appendix H, as well as in the extended related works discussion in Appendix J.
>
> --
>
> [1] “Context matters for image descriptions for accessibility: Challenges for referenceless evaluation metrics,” Kreiss et al. 2022
>
> [2] “An Explanation of In-context Learning as Implicit Bayesian Inference” Xie et al. 2021
>
> [3] “Rethinking the Role of Demonstrations: What Makes In-Context Learning Work?” Min et al. 2022

---

> > ### Author Response · Authors · 2023-11-22
> >
> > We again thank the reviewer for their comments and suggestions, and we hope to have addressed them in the revised paper we uploaded and the individual responses. If there are any remaining open questions or concerns, please let us know and we would be happy to discuss them and further revise our manuscript accordingly.

---

### Official Review · Reviewer_hp9w · 2023-10-31

**Soundness:** 3 good
**Presentation:** 2 fair
**Contribution:** 3 good
**Rating:** 6
**Confidence:** 3

**Summary:**

This paper proposes an evaluation protocol to evaluate referenceless metrics commonly used for the assessment of text image description generators. Authors first build an evaluation dataset as a subset of the WIT dataset comprising 204 image-caption-context-description samples. This dataset is used for a user study, where humans are rating the quality of different image description across multiple axes. Human preference is then used as a reference to evaluate the quality of referenceless metric, alongside measures of metric robustness by applying several transformations to the generated descriptions. Finally, authors investigate whether model-based metrics can be improved by fine-tuning models on human scores and augmentations.

**Strengths:**

The paper addresses an interesting topic. Referenceless metrics are often used as a means to demonstrate the performance of a new model, with limited insights into how reliable they can be. Providing a solution to understand which metrics are most accurate has the potential to be very useful.

Authors have evaluated a large set of model-based evaluation metrics, with the somewhat low correlations suggesting a lack of reliability. Furthermore, the reference dataset and associated human study are a useful resource that could be used to evaluate future metrics.

**Weaknesses:**

The paper presentation could be substantially improved. The related work section is focusing almost solely on the clip model,and fails to provide an accurate review of pre-existing related literature. Authors should review pre-existing metrics, related benchmarks and evaluation strategies. Similarly, section 3 provides descriptions of metrics being analysed in this work, and lacks accurate descriptions to understand the particularities of each method. For example, only encoder architectures are discussed for the Flamingo model, but the actual mechanisms used to compute the metric are not discussed. The strategy used to integrate context information as also poorly explained, and could benefit from providing equations.

Certain claims seem excessive. Authors mention that their results suggests that referenceless metrics can be successfully used based on observed correlation, however somewhat low correlations are observed (.2-.3), suggesting the opposite. In addition, the use of context is highlighted multiple times throughout the paper, yet the impact of integrating this context variable is barely discussed, the only relevant experiment is the use of shuffled contexts in section 5.2. It would have been interesting to discuss context more in depth and how it impacts user evaluation and metric performance.

The fine-tuning experiment in section 6 doesn’t seem particularly necessary, as metrics become tailored for a specific dataset/user preference, This reduces applicability and generalizability, and seem to contradict the main objective of the paper.

**Questions:**

- Section 4.1 mentions that the dataset is built by collecting caption-alt description pairs for each image, yet in the human study, it is mentioned that 5 descriptions are provided. Where are the additional descriptions obtained from?

- There appears the be a lot of similarities between the data collection and human study protocol proposed in this paper and in Kreiss et al 2022b. Can authors clarify how related these two strategies are?

---

> ### Author Response · Authors · 2023-11-17
>
> We thank the reviewer for all of the constructive comments and helpful suggestions. We updated the paper accordingly, and summarize our changes and responses here. We hope that they address the reviewer's concerns; however, if there are specific areas where we can further improve, please don't hesitate to let us know.
>
> *> The related work section is focusing almost solely on the clip model,and fails to provide an accurate review of pre-existing related literature. Authors should review pre-existing metrics, related benchmarks and evaluation strategies.*
>
> We agree that the paper benefits from discussing those related works in more detail and updated our paper accordingly. Since we had to contend with the space limitations, we opted for discussing much of the related literature in the thematically corresponding sections wherever possible (Section 2 for referenceless metrics and robustness checks, Section 3 for relevant models, Section 4 for evaluation frameworks and annotation guides, and Section 5 for common model-generated errors) and specifically elaborate on pre-existing metrics as well as prior evaluation strategies and benchmarks in a new Appendix section (Appendix I).
>
> *> Section 3 provides descriptions of metrics being analysed in this work, and lacks accurate descriptions to understand the particularities of each method. For example, only encoder architectures are discussed for the Flamingo model, but the actual mechanisms used to compute the metric are not discussed. The strategy used to integrate context information is also poorly explained, and could benefit from providing equations.*
>
> Thanks for pointing this out! We’ve added some equations to clarify how context is incorporated, we streamlined the model details provided in Section 3 and added more model details in Appendix H. For the likelihood-based approaches, context integration is simply accomplished by including the context as part of the input text when computing likelihood. There are certainly other ways of incorporating context into these likelihood-based models, but we leave this to future work. We’ve also reproduced the equation from Kreiss et al. (2022) to help clarify the role of context in the tested CLIPScore.
>
> *> Authors mention that their results suggests that referenceless metrics can be successfully used based on observed correlation, however somewhat low correlations are observed (.2-.3), suggesting the opposite.*
>
> We agree that the previous version of the paper failed to properly contextualize the size of the correlations and we are aiming to address this here. Due to the interannotator variation, the highest correlation that is theoretically achievable on our dataset is actually much lower than 1. Depending on the question, it ranges between 0.58 and 0.66. (We computed this by correlating our dataset with the average rating for each description.) However, this still highly overestimates the realistic upper bound for a correlation. To address this, we created a counterfactual dataset by sampling with replacement from the original dataset and correlated the average ratings of the counterfactual dataset with the original data. This suggests a more realistic human upper bound with our data at between 0.42 and 0.57 depending on the question. These numbers are further supported by prior work correlating different participant populations on description quality tasks. Kreiss et al. (2022) found a correlation of about 0.5 when comparing sighted and BLV user judgments. These prior observations better contextualize the quality of the correlations reported in the paper. We agree that correlations below 0.3 should not be considered strong and we agree with the reviewer that our paper doesn’t rest on the assumption that those correlations are particularly high. In the updated paper, we adjust our language accordingly. However, for post-image ratings on the overall question, 5 out of the 7 metrics achieve a correlation higher than 0.35 and we believe that the estimates of an upper correlational limit suggest that correlations of 0.35 and above are very impressive, given the interannotator variation that is a core characteristic of psychological preference studies. We also added further discussion on the interannotator variation in Appendix H. We integrated the estimated correlation ceiling into Figure 2 in the updated paper to help readers contextualize the size of the correlation.
>
> (We also realized that the y axis breaks in Figure 2 weren’t well chosen to identify the correlation values. We therefore changed y axis breaks in Figure 2 to make the values clearer and added a table with the exact correlation values to the appendix.)

---

> > ### Author Response · Authors · 2023-11-17
> >
> > *> In addition, the use of context is highlighted multiple times throughout the paper, yet the impact of integrating this context variable is barely discussed, the only relevant experiment is the use of shuffled contexts in section 5.2. It would have been interesting to discuss context more in depth and how it impacts user evaluation and metric performance.*
> >
> > The reviewer is correct in that the only explicit context manipulation we use is the shuffled context augmentation. However, context plays an important implicit role within the whole framework in the following two ways. Most crucially, against common practice in image description annotation today, we present the images and descriptions within the context they occur. This means that ratings are provided in relation to contextually available information and (as shown by Kreiss et al. 2022b) this will affect the correlation analyses (Figure 2). The relevance and imaginability questions tap into more and less context-sensitive quality measures, and can therefore function as diagnostic tools for a metric’s contextual sensitivity. Secondly, context is part of how all metrics in this paper are computed. It’s true that we’re not explicitly comparing performance of no-context vs. with-context metrics, because (A) we see it as evident from prior work (e.g., Stangl et al. 2020, Stangl et al. 2021, Muehlbradt et al. 2022) that context needs to play a fundamental role in description quality judgments, and (B) we chose to focus on establishing the framework itself, and our augmentation and finetuning experiments are already sufficient to uncover remaining challenges with effective context integration.
> >
> > We agree that there is more work to be done about explicitly exploring the contextual capabilities of the models themselves. With this paper, we aim to introduce the framework to do that and we’re looking to apply this framework for a detailed investigation of context-sensitivity in today’s models in future work.
> >
> > –
> >
> > Abigale Stangl, Meredith Ringel Morris, and Danna Gurari. "Person, shoes, tree. Is the person naked?" What people with vision impairments want in image descriptions. Proceedings of the 2020 CHI Conference on Human Factors in Computing Systems, 2020.
> >
> > Abigale Stangl, Nitin Verma, Kenneth R Fleischmann, Meredith Ringel Morris, and Danna Gurari. Going beyond one-size-fits-all image descriptions to satisfy the information wants of people who are blind or have low vision. In
> > Proceedings of the 23rd International ACM SIGACCESS Conference on Computers and Accessibility, 2021.
> >
> > Annika Muehlbradt and Shaun K Kane. What’s in an alt tag? exploring caption content priorities through collaborative captioning. ACM Transactions on Accessible Computing (TACCESS), 2022.
> >
> > –
> >
> > *> The fine-tuning experiment in section 6 doesn’t seem particularly necessary, as metrics become tailored for a specific dataset/user preference, This reduces applicability and generalizability, and seem to contradict the main objective of the paper.*
> >
> > Thank you for raising this point. The goal of this section is to consider a way forward, motivated by recent trends in research on large pretrained models. Our analyses suggest that even the sophisticated models we can currently leverage can’t effectively assist us in guiding image description quality judgments. However, they still have promising characteristics, like access to a vast amount of world knowledge and a general understanding of semantic similarities (the picture above the table vs. the table below the picture) that still makes them promising resources for description evaluation. Recently, due to similar limitations, techniques such as instruction tuning and reinforcement learning from human feedback (RLHF) have been used to help close gaps between the capabilities of large pretrained models and real world usage (e.g. Ziegler et al. 2019’s “Fine-Tuning Language Models from Human Preferences”). In Section 6, we provide a case study to show that with specific goals in mind, finetuning can provide a promising strategy to start counteracting some of the undesirable behavior. However, this doesn’t alleviate the general purpose and findings of the previous sections and based on your feedback, we highlighted that in the new paper draft to make the goal of Section 6 clearer.

---

> > > ### Author Response · Authors · 2023-11-17
> > >
> > > *> Section 4.1 mentions that the dataset is built by collecting caption-alt description pairs for each image, yet in the human study, it is mentioned that 5 descriptions are provided. Where are the additional descriptions obtained from?*
> > >
> > > Thank you for bringing this to our attention – there was indeed an ambiguity in the way we described the experimental setup. Each participant saw five images within their respective contexts, each associated with a single description which they rated. We clarified this now in the paper (Section 4.2).
> > >
> > > *> There appears to be a lot of similarities between the data collection and human study protocol proposed in this paper and in Kreiss et al 2022b. Can authors clarify how related these two strategies are?*
> > >
> > > We largely adopt the Kreiss et al. (2022b) study protocol because it focuses on a goal-driven framing that allows us to investigate inherent communicative preferences instead of adherence to training and instructions, which can induce its own biases. This annotation framework was further established to correlate well with Blind and Low-Vision (BLV) users and the design explicitly incorporates contextual information, which is crucial to our study.  Kreiss et al. (2022b) introduce this protocol, but only test it on data that was artificially constructed, i.e., the authors paired each image by-hand with a variety of contexts. We use naturalistic data with realistic contexts that are also more complex (e.g., in contrast to the prior setup, we include the caption of the image and a closely appearing context paragraph). Furthermore, we added the “Added Info” question because in the naturalistic data, the caption and description were sometimes identical, and we wanted to gain better understanding on how contextual redundancy affects rater behavior. The result is a first naturalistic dataset that is labeled following a similar methodology that was introduced by Kreiss et al. (2022b).

---

> > > > ### Author Response · Authors · 2023-11-22
> > > >
> > > > We again thank the reviewer for their comments and suggestions, and we hope to have addressed them in the revised paper we uploaded and the individual responses. If there are any remaining open questions or concerns, please let us know and we would be happy to discuss them and further revise our manuscript accordingly.

---

### Official Review · Reviewer_jrgb · 2023-10-31

**Soundness:** 2 fair
**Presentation:** 2 fair
**Contribution:** 2 fair
**Rating:** 6
**Confidence:** 1

**Summary:**

This work introduces a new benchmark to evaluate the alignment of referenceless metrics, like CLIPScore, with human preferences in the context of image description generation. It comprises human ratings and robustness checks, emphasizing the importance of context in image descriptions. The assessment reveals that none of the existing methods fully meet the benchmark, especially in context sensitivity. However, the paper suggests that fine-tuning can enhance metric performance, although ContextRef remains a challenging benchmark due to the complexity of context dependence.

**Strengths:**

ContextRef represents a significant advancement in the evaluation of referenceless metrics for image description generation. By emphasizing the importance of context, it addresses a critical gap in existing benchmarks.
The benchmark includes both human ratings along various quality dimensions and robustness checks. This comprehensive approach allows for a more nuanced understanding of the strengths and weaknesses of referenceless metrics.
By focusing on context, ContextRef aligns more closely with real-world scenarios where image descriptions are typically encountered, making the benchmark more relevant and applicable.
he paper sheds light on the often-underestimated role of context in image description quality, contributing valuable insights to the field and encouraging future research to consider this aspect more thoroughly.

**Weaknesses:**

This work involves human ratings, which are subjective and can introduce bias. The paper could benefit from a more detailed discussion of how raters were selected and trained to ensure consistency and mitigate potential biases.

Overall, I am not working on this area at all, I would like to check comments from other reviewers for the final rating.

**Questions:**

no.

---

> ### Author Response · Authors · 2023-11-17
>
> We thank the reviewer for their thoughtful engagement with our paper and for raising the following important point, which we have now addressed in the paper.
>
> *> This work involves human ratings, which are subjective and can introduce bias. The paper could benefit from a more detailed discussion of how raters were selected and trained to ensure consistency and mitigate potential biases.*
>
> To address this point, we added a subsection with details on the participant population (Appendix A) and we added a detailed limitations section (Appendix I). For convenience, we summarize our response here, as well.
>
> The goal of the ContextRef framework is to test model-based metric alignment with people’s intuitions for description quality for an accessibility goal. However, as the reviewer rightfully points out, there is interannotator variance, and people’s cultural backgrounds and individual needs necessarily shape their preferences. To get first traction on the core problem itself, we get multiple ratings for each individual description, which reduces the effect of outliers but reflects general preference patterns in the recruited population. This is not to say that outliers aren’t meaningful, and we added a discussion on two core sources of variance.
>
> Firstly, we elaborate on the annotator selection and discuss the implications of it. This includes that recruitment was restricted to US-based adults who are part of the Prolific platform. We agree that this leads to implicit biases in the data that might not generalize to participant populations outside of our recruitment parameters, which is a clear limitation that we now elaborate on in the paper. In the paper, we focus on a pipeline that allows us to get initial traction on the overall problem of using models for image description quality assessment, but this doesn’t yet allow insights on important perspectives such as cultural representation and equity promoted by these models, which need to complement our framework.
>
> Secondly and in addition to cultural variation, there is individual variation, which is an important component of current image description research. One such example is description length. A variety of papers have found conflicting evidence on whether shorter or longer descriptions are generally preferred (Slatin & Rush, 2003; Petrie et al., 2005; Rodríguez Vázquez et al., 2014). We therefore chose to set up this study based on a goal-oriented framing (consider the purpose and choose your responses accordingly) as opposed to explicit training and instructions. However, we encourage work that builds on ContextRef to adjust the experimental design according to potentially distinct downstream needs. To incorporate the individual uncertainty into our framework, we correlate metric predictions to individual rating distributions instead of the averaged ratings for each description.
>
> Finally, we would like to emphasize that we chose this experimental method since it was established by prior work with sighted as well as Blind and Low-Vision (BLV) participants, which helps ground our findings in the existing literature.
>
> We hope that this addresses the reviewer’s concerns but if there are any particular points, we can expand on, please let us know.
>
> –
>
> John M Slatin and Sharron Rush. Maximum accessibility: Making your web site more usable for everyone. Addison-Wesley Professional, 2003.
>
> Helen Petrie, Chandra Harrison, and Sundeep Dev. Describing images on the web: a survey of current practice and prospects for the future. Proceedings of Human Computer Interaction International (HCII), 71(2), 2005.
>
> Silvia Rodríguez Vázquez, A Bolfing, and P Bouillon. Applying accessibility-oriented controlled language (cl) rules to improve appropriateness of text alternatives for images: an exploratory study. Proceedings of LREC 2014, 2014.

---

> > ### Author Response · Authors · 2023-11-22
> >
> > We again thank the reviewer for their comments and suggestions, and we hope to have addressed them in the revised paper we uploaded and the individual responses. If there are any remaining open questions or concerns, please let us know and we would be happy to discuss them and further revise our manuscript accordingly.

---

### Author Response · Authors · 2023-11-23

We thank all reviewers for their thoughtful comments and helpful suggestions. We appreciate that reviewers found our work to be a "significant advancement" in measuring the effectiveness of referenceless image description evaluations (**Reviewer jrgb**), and felt that it has the potential to be "very useful" (**Reviewer hp9w**). We also appreciate that our efforts to critically examine and improve these models were considered a "valuable contribution to the field" (**Reviewer T1dS**).

Based on the broadly supportive feedback and in response to the helpful suggestions, we made several revisions to the paper and added multiple contextualizing analyses, including clarifying our methodology and expanding our discussions on the limitations and potential biases of our dataset.

The main changes include:

1. **Expanded Discussion on Human Ratings and Bias** (Appendices A, I, and J): In response to questions about the subjectivity and potential bias of human ratings (**Reviewer jrgb**), we have added detailed information on the selection and training of raters. We also discuss the limitations of our approach and how it might affect the generalizability of our findings.

2. **Clarified Descriptions of Analyzed Metrics** (Section 3, Appendices H and J): **Reviewers hp9w and T1dS** requested additional information about the exact way in which the scores are computed and context is integrated. We have included detailed explanations and equations to better describe the metrics in our work. Depending on the underlying model, this includes the exact model prompts, equations that specify context integration, and the actual computation of the scores.

3. **Contextualizing Correlation Values** (Appendix H and Figure 2): Following feedback about the correlation values observed in our study (**Reviewer hp9w**), we have provided additional context to better understand these values in light of interannotator variation, reformatted the main figure and provided an additional results table in the appendix, and have adjusted our language to reflect the realistic upper bounds of these correlations.

4. **Related Literature** (Appendix I, etc.): To address concerns about the focus on the CLIP model and the need for a broader literature review (**Reviewer hp9w**), we added additional discussion of pre-existing metrics, benchmarks, and evaluation strategies in Appendix I. We also clarified that we discuss additional related works in their respective sections (Section 2 for referenceless metrics and robustness checks, Section 3 for relevant models, Section 4 for evaluation frameworks and annotation guides, and Section 5 for common model-generated errors).

Again, we are grateful to all the reviewers for the valuable points raised, and believe that our paper has been significantly improved as a consequence. We believe that our revisions and responses have addressed all the concerns raised by the reviewers, and we look forward to further comments and suggestions.

---

### Meta-Review · Area_Chair_qaVN · 2023-12-24

**Metareview:**

This paper evaluates different "referenceless" metrics for image caption generation (where reference means without ground truth caption), presenting the descriptions in context. The paper assesses different existing metrics (like CLIP, BLIP) under different conditions and methods of incorporating context. The paper finds that current metrics do not reflect human preferences for incorporating context. The paper also assesses how different data augmentations, which should decrease the score, affect the actual outcome. The paper finds several metrics are not robust to such augmentations. Overall, the reviewers appreciated the contribution of the benchmark, and all agree the paper is above the bar for acceptance. The authors submitted a rebuttal, especially for comments from Reviewer hp9w, who wrote the most thorough review. The revewers did not respond, but the AC agrees that the paper is a solid contribution and agrees with the acceptance rating.

**Justification For Why Not Higher Score:**

The paper is a solid contribution for evaluating benchmarks for image captioning. No reviewer championed a spotlight or oral.

**Justification For Why Not Lower Score:**

All reviewers agree the paper is above the bar for acceptance and the AC agrees.

---

### Decision · Program_Chairs · 2024-01-16

Accept (poster)